# Improving Deep Learning for Maritime Remote Sensing Through Data Augmentation and Latent Space

Daniel Sobien [1,*], Erik Higgins [2], Justin Krometis [1], Justin Kauffman [1] and Laura Freeman [1]

1   National Security Institute, Virginia Tech, Arlington, VA 22203, USA; jkrometi@vt.edu (J.K.); jakauff@vt.edu (J.K.); laura.freeman@vt.edu (L.F.)
2   Department of Aerospace and Ocean Engineering, Virginia Tech, Blacksburg, VA 24061, USA; erikth1@vt.edu
*   Correspondence: sdan8@vt.edu

**Abstract:** Training deep learning models requires having the right data for the problem and understanding both your data and the models' performance on that data. Training deep learning models is difficult when data are limited, so in this paper, we seek to answer the following question: how can we train a deep learning model to increase its performance on a targeted area with limited data? We do this by applying rotation data augmentations to a simulated synthetic aperture radar (SAR) image dataset. We use the Uniform Manifold Approximation and Projection (UMAP) dimensionality reduction technique to understand the effects of augmentations on the data in latent space. Using this latent space representation, we can understand the data and choose specific training samples aimed at boosting model performance in targeted under-performing regions without the need to increase training set sizes. Results show that using latent space to choose training data significantly improves model performance in some cases; however, there are other cases where no improvements are made. We show that linking patterns in latent space is a possible predictor of model performance, but results require some experimentation and domain knowledge to determine the best options.

**Keywords:** data augmentation; dimensionality reduction; latent space; UMAP; simulated data; deep neural network; synthetic aperture radar

## 1. Introduction

Understanding your data is the first step for any deep learning model because it informs not only how to solve the problem and what architecture to use but also how to train and test the model. Specifically, in this paper, we investigate the following question: "how can we train a deep learning model to increase its performance on a targeted region where data are limited"? To answer this question, we study the effect that data augmentations, specifically image rotation, have on a small dataset of simulated synthetic aperture radar (SAR) imagery.

Our goal is not only applying data augmentations to generalize a deep learning model and understand the simulated data but manipulating the data in a meaningful way to improve performance in a targeted area. This study provides a foundation to bridge the gap between using simulated data and real-world data when training a deep learning model. Real-world data are difficult to obtain, so simulated data are used as a surrogate for our application of maritime remote sensing. The simulated data are created via high-performance computing models that simulate the hydrodynamics of a surface ship's wake and the resulting SAR imagery of the sea surface. These computational models are costly in both time and computing resources, where some cases can take days to run. Data augmentations, however, can generate "new" images in a time- and cost-effective manner.

Using latent space representations, we analyze relationships among SAR images for generative characteristics and assess the effects that augmentations have on the data domain. The motivation for using latent space is to visualize not only the relationship among the

images but to see how the augmentations change this relationship, specifically to see how augmentations move the images within the latent space. If we can relate training data in latent space for a model, we can predict how well the model is expected to perform on test data in the latent space. For example, if we observe that data from different SAR frequency bands overlap in latent space, we expect a single model trained on only one band to perform well on all bands of the data in the overlapped region. Likewise, if data are augmented to be near a targeted area, we can choose the training data to improve performance.

We assess the effects of rotation augmentations on the models by using sensitivity analysis to identify the region of lowest performance. To create the latent space representations of both the original and augmented data, we use dimensionality reduction via Uniform Manifold Approximation and Projection (UMAP). The latent space representation informs both a baseline study to assess how the UMAP latent space is related to model performance and a performance study to see how choosing augmented data from the latent space can improve model performance outside the original operating envelope of the model.

A key contribution of this work is a process for understanding the effect that augmentations have on both data and model performance. The goal is to use that understanding to choose training data and improve model performance. To the best of our knowledge, there is no prior research on the direct use of augmentations to target a specific area of improving model performance. Our contribution is to show how latent space allows us to see the effects of data augmentations and to then inform which augmentations are best to use for model performance. This contribution informs those using simulated and/or small datasets how best to employ data augmentations in a more effective manner than randomly augmenting images. While we have not used real-world data for this study, this work is foundational for relating simulated data to real-world data via latent space representation and then employing data augmentations to bridge the gap from the simulated data domain to the real-world data domain.

## 2. Background

### 2.1. What Are Data Augmentations?

Data augmentations are known to have a positive effect on deep learning models [1]. The concept is simple: augmentations add variety and randomness to data, generalizing a deep learning model to perform well on new data by not overfitting on the training data. The wake identification task is a classification problem, so augmentations must not alter the data enough to change the label [2].

Beyond this understanding, there has been less work looking into specific effects of data augmentations to the training process of deep learning models. For instance, the most basic application of augmentations is to increase dataset size. Perez and Wang [3] applied a random augmentation to each image in a dataset of size $N$ to produce a dataset of size $2N$, and a survey from Shorten and Khoshgoftaar [1] presented similar findings for the deep learning field. Simply applying an augmentation to an image seems to be a justification for calling the result a new image, provided we follow the rule from Goodfellow et al. [2] that we do not alter the image enough to change the label.

Tran et al. [4], however, questioned this approach of "simple geometric and appearance transformations" to existing images. They also stated that random augmentation applied to images, while still preserving the label, "assumes that the noise model over these transformation spaces can represent with fidelity the processes that have produced the labeled images" and that such assumptions have not been properly tested [4].

While our goal is not to directly test these assumptions, they do present some limitations when applying augmentations to any data. Augmentations become more difficult when looking at physics-based imagery, such as our simulated SAR images of ship wakes. There is a smaller window for augmenting the data without altering its physical meaning. Fabian et al. [5] expressed similar concern when augmenting accelerated Magnetic Resonance Imaging (MRI) images, where any augmentation applied to the data had to account for the physics of the MRI process. Our approach for preserving the physical meaning

is applying domain expertise to the problem and focusing on the rotation augmentation, which has little effect on the SAR physics.

When it comes to SAR imagery, there are considerations to take into account that are not present in electro-optical (EO) imagery, such as noise, self-shadowing, and dynamic range, all of which impact the image depending on the direction of illumination, meaning that augmentations will not impact SAR in the same way as EO images [6]. This means there are assumptions made when augmentations are applied to the SAR images; namely, that the rotation augmentation featured in this paper is rotating the sensor along with the wake to preserve the orientation between sensor and wake. While with EO imagery, we could assume that the wake is orientated different to the observer, we cannot make the same assumption for SAR imagery.

This does not diminish the effectiveness of data augmentations for SAR imagery, and in fact this has been a well researched topic for SAR automatic target recognition (ATR) [7–10]. This previous research, however, focuses on the SAR of objects, whereas our application focuses on the SAR imagery of the persistent ocean wakes that are a result of these objects. Ding et al. [7] used a simple convolutional neural network (CNN) model and applied a rotation augmentation to an objects SAR image via a linear combination of two similar images, which they called pose synthesis. Du et al. [9], however, applied rotations directly to the images themselves when training their CNN model. Other relevant data augmentations to SAR imagery include translation or displacement [7,9], speckle noise [7,8], and attributed scattering centers (ASC) [10].

This last technique, ASC, relied on cleaning and improving the SAR image to only the object's characteristics so it could be used to generate new SAR images for training, as the augmented set "covers more operating conditions in SAR... therefore, the trained CNN by the augmented samples could better handle these nuisance conditions" [10]. In fact, gathering real-world data of relevant SAR imagery is difficult, and this has been the motivation for much of the previous work discussed above. This difficulty has also motivated the use of simulated SAR data in place of real-world data for training CNN models [11,12]. Malmgren-Hansen et al. [11] used physics-based model (Computer Simulation Technology (CST) Microwave Studio Asymptotic Solver) and 3D computer aided design (CAD) models to generate SAR images of objects. Wang et al. [12], however, adopted the deep learning approach and used generative adversarial networks (GANs) to create simulated SAR images. Both these studies used transfer learning to first train a model on simulated data and then fine-tune the model on real-world data [11,12].

### 2.2. Dimensionality Reduction Techniques

Common dimensionality reductions techniques include Principal Component Analysis (PCA), T-distributed Stochastic Neighbor Embedding (t-SNE), and Uniform Manifold Approximation and Projection (UMAP). Each have their own advantages and disadvantages for this application. PCA is a common and established method that reduces the dimensionality of data while maximizing variance in the resulting dimensions by solving an eigenvector problem [13]. PCA, however, is a linear technique and can result in overcrowding where clusters fail to form distinct groups that are easy to identify [14]. t-SNE is an iterative technique using manifold learning that is better at preserving structure at several scales, local (i.e., small-scale or intra-cluster features) and global (i.e., large-scale or inter-cluster features), in the resulting clusters [14,15]. A drawback of t-SNE, however, is the slow computational speed [14], which was the major reason for not using t-SNE on our dataset. UMAP is another manifold learning technique, which claims to preserve more global structure than t-SNE with better run-time performance [14,16], and for both of those reasons we chose UMAP.

### 2.3. Ship Wake SAR Imagery Dataset

The SAR images are generated using end-to-end physics-based modeling and simulation tools developed to model complex phenomena on the ocean surface. The simulations

are described by Higgins et al. [17] but are summarized here: computational fluid dynamics (CFD) simulations model the evolution of a surface ship wake in a section of ocean; then, a second hydrodynamic simulation redistributes surfactants according to the surface currents and turbulence within the ship wake. More details about the hydrodynamic simulations that produced these SAR images can be found in [17], including the specific ship parameters, wind, ship speed, and headings. Altering these parameters will impact the images generated and provide a larger data set, but this is currently beyond the scope of the work presented here. This work focuses on understanding how data changes when augmentations are performed and how we can more effectively and efficiently grow our data set.

Hydrodynamic data, including surfactant concentration, are extracted and used as an input to the Environmental Research Institute of Michigan (ERIM) Ocean Model [18], which calculates the surface wave spectrum and simulates the SAR cross section of the ocean surface, including the region that contains the ship wake. These radar cross section data are then used to construct the simulated SAR images. Variables including swell height and SAR platform parameters (see Table 1 from Higgins et al. [17]), such as look angle and polarization, are systematically varied between the hydrodynamic and electromagnetic simulations to create realistic variation in the SAR images. The advantage of simulation is the ability to control these variables, which is especially helpful for studying the effects of data augmentation before applying it to real-world data, where there are more unknowns.

**Table 1.** Run matrix for SAR image generation from [17]. The look angle is the azimuth angle between the radar direction and the ship direction, and the inclination angle is the angle between the vertical axis and the direction from the radar to the ship wake [18].

| Parameter | Possible Values |
| --- | --- |
| SAR band | C, S, X |
| Look angle | 0°, 90°, 180°, 270° |
| Polarization | VV, HH |
| Inclination angle | 30°, 40°, 50°, 60° |

Typically, look angles (the azimuth angle between the radar direction and the ship direction) are collected in a way that only uses 90 and 270 angles (i.e., looking either left or right). In application, look angles would be limited to a more physically realizable range depending on the specific SAR system in use; however, this study is not intended to be descriptive of any particular system but rather focuses on an abstract framework. We are interested in simulating data for the entire parameter space. This was done to create as large a domain for the deep learning algorithms as possible and demonstrate what insights can be gained. Wang et al. [19] simulated SAR images of oceanic shear-wave-generated eddies using 0°, 90°, 180°, and 270° degree look angles and found that features vary between the 90° and 270° look angle images and the 0° and 180° look angle images. This is not the same phenomena we are modeling, but we found similar differences in SAR wake images. There are features that only exist in 0° and 180° look angles, and even if we do not expect to see them in real-world data, using them should result in a model that is invariant to the look angle and therefore more robust.

Along with the SAR images are ground truth wake segmentation masks that mark the location of the wake from the hydrodynamic simulation. The masks correspond to the hydrodynamic simulation only, so multiple SAR images generated from the same case share the same wake mask. Figure 1 shows an example of a simulation with a persistent wake (top row) where the different SAR band images share the same mask on the far right.

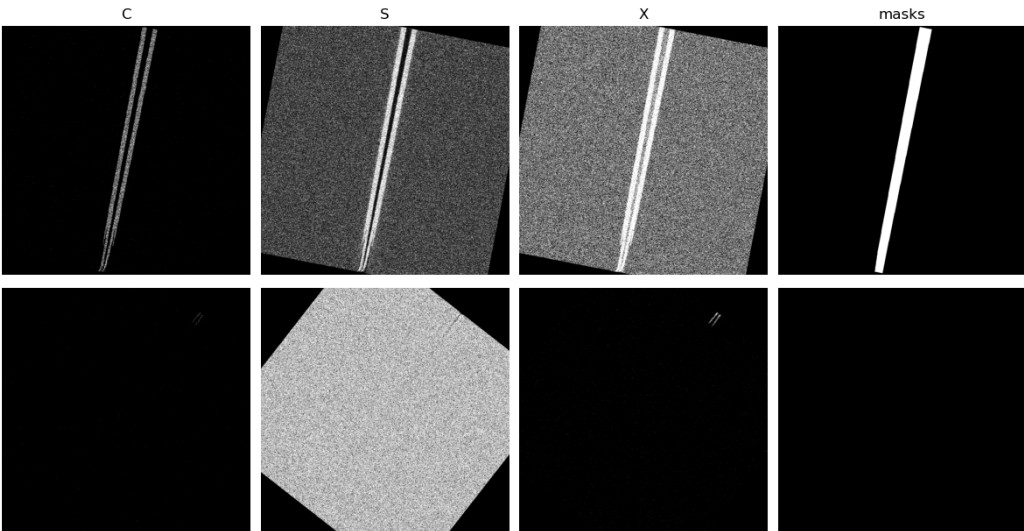

**Figure 1.** Images for two wake events (one per row) are displayed in this figure. The first column on the left is the C-band images, followed by the S-band and X-band images. The final column to the right is the wake masks used for the U-Net segmentation model. Note that there is no persistent wake for the images in the second row, and therefore there is no mask. All images with a wake event have the same augmentation applied to them for consistency in the augmented dataset.

### 2.4. Deep Learning Models for Ship Wake Detection

In previous work [17], we developed three convolutional neural network (CNN) models for classifying the presence of a persistent wake in SAR imagery. Previous work has shown the effectiveness of CNN models for SAR imagery [7–12]. There is a single baseline model, referred to as the "base" model, which consists of three convolutional layers and two fully connected layers. The other two models build on this by allowing additional information to be passed into the model, but the remaining CNN architecture (the three convolutional layers and two fully connected layers) remains the same. The other two models are "unet", which incorporates a U-Net segmentation model, based on Ronneberger et al. [20] and implemented by Usuyama [21], into the base CNN architecture, and "clss", which duplicates some of the input channels to match the input data size of the unet model but only uses the base CNN architecture. In [22], we found the unet model outperforms the base and clss models, so we focus on the unet model in this paper; however, some results are presented for all models.

The unet model creates a segmentation mask that highlights the wake region as a form of wake feature extraction. This process required training the U-Net segmentation model first and then passing the output segmentation mask into the CNN classifier, which concatenates the mask with the raw SAR image, making the U-Net segmentation mask another channel of the input image.

## 3. Methods

### 3.1. How We Augmented the Data

We applied the data augmentations using utilities from the PyTorch library [23] but developed custom interfaces to apply the augmentation function in exactly the same manner as the input wake images and the associated ground truth mask. This ensures the labeled data for the U-Net segmentation model were augmented in the same way as the input image.

Augmented sets were created by applying the same augmentation (e.g., rotate by 45°) to the original image dataset and saving the results in separate directories. This ensured that the augmented images matched for every test, making the results of the studies in this paper more directly comparable. The input SAR images and masks were rotated by a series

of fixed angles from 0° to 180° in 15° increments. Within each augmentation set were three sub-directories, one for each SAR band.

Figure 1 presents two wakes with the rotation augmentation. Each row is a single wake event from a different augmentation set, and the first three columns from the left are the C-, S-, then X-band SAR image of the wake. The last column on the right is the wake mask used for the ground truth of the U-Net segmentation model.

The rotation augmentation refers to the geometry of the image and not the look angle of the SAR simulation. Similar rotation augmentations were applied by Du et al. [9]. SAR imagery is impacted by several different factors given that the sensing is active versus the passive sensing of EO cameras [6]. Therefore, the rotations applied to the SAR imagery assume that the sensor rotates with the wake, preserving the orientation between sensor and wake. This preserves the physics of the data, while allowing the deep learning model to see wakes at new orientations. Although this is not the same as the sensors seeing wakes at different orientations, it is a simple way for us to teach the model to look for wakes at other orientations. Otherwise, the horizontal direction of the wake would be overrepresented in the original dataset, and the deep learning model would have difficulty on any rotated wake image, as we see in Figure 2.

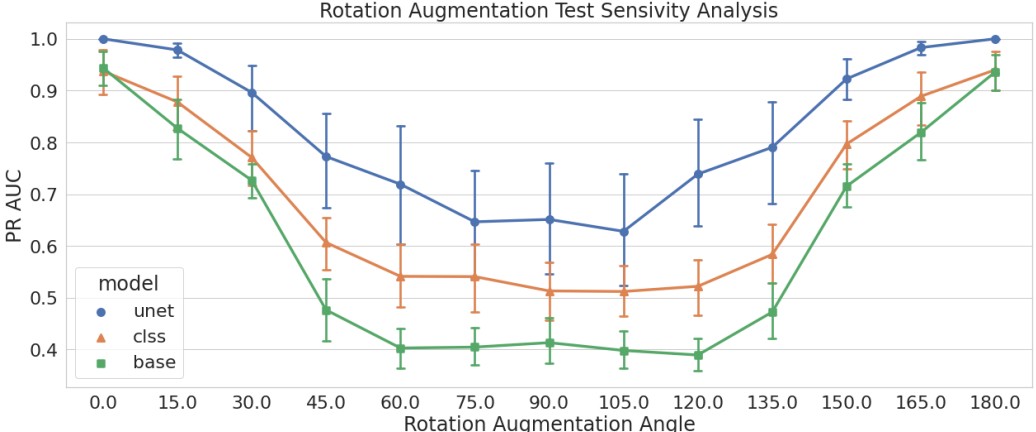

**Figure 2.** Performance of models trained with no augmentations tested on all rotation augmentations sets. Models are measured using the PR AUC metric, and we see models drop to their lowest performance around the 90° rotation.

### 3.2. Performing a Sensitivity Analysis

The sensitivity analysis in this paper is straightforward: a model is trained with no augmentations (this referred to as the naïve model) and is then tested on a series of augmented datasets to see how performance changes with augmented data. The area under the precision-recall curve (PR AUC) is the metric of choice because it does not require us to set a specific threshold, and it can handle imbalanced data better than a Receiver Operating Characteristic (ROC) curve [24,25]. While great for a quick snapshot and comparison of different models, it can give slightly unrealistic expectations of a model deployed in real-world cases, so the absolute values of the metrics must be interpreted with caution. The PR AUC metric is calculated using the `precision_recall_curve` and `auc` functions from [26], but the equation can be expressed in terms of precision, *P*, and recall, *R*, at each threshold, *i*, as

$$PR\,AUC = \frac{1}{2}\sum_{i=1}^{n}(P_{i-1}+P_i)(R_i-R_{i-1});\ i=\{0,1,...,n\},$$

where thresholds are a set of values from 0 to 1 that determine the cutoff for wake present (if model predicts value above threshold) or no wake present (if the model predicts a value below the threshold). Threshold values are automatically determined by the function `precision_recall_curve` from [26].

We test the naive models with the fixed rotation augmentation sets from 0° to 180°. We expect to see the performance drop as the rotation angle moves away from 0° and expect the performance on some sets to return as the angle gets close to 180° because the wake returns to a horizontal position in the image.

There are several different naive models trained for the sensitivity analysis based on both model architecture and SAR bands of the training data. Three model architectures (unet, clss, base) and four combinations of SAR bands (C, S, X, and CSX) are used for a total of 12 models. The combined CSX images use each band as a different channel in the image, just like a color image uses red, blue, and green channels of visible light; however, here we use C-, S-, and X-band channels of the radio frequency spectrum. There are five iterations of training and testing for each model to build a more statistically meaningful set of results and help curb any effects of randomness for any single model. Some results, however, still show notable variance, but the trends in the results are the important finding. Because we are interested in the overall effect of the augmentations, the results are presented for each architecture separately but aggregated over all the SAR band versions of the models.

The results for the rotation augmentation sensitivity analysis are presented in Figure 2. The performance drops quickly as the rotation angle moves away from 0°. The baseline model performance drops by approximately 20% by the time it hits 30° and quickly sinks after that. We see less effect on the unet and clss models but more variance (uncertainty) in those results. The model performance increases when the angles get close to 180°, as the wakes start to return to a horizontal position in the images.

The 90° rotation region (from 75° to 105°) marks the worst performance for all the models, which is not surprising considering that 90° is the greatest change in the original image that we can create via rotations. This is the region, specifically at 90°, in which we are most interested in improving model performance.

### 3.3. Creating the Latent Space

In this section, we explore reducing images into a lower dimensional latent space and how the rotation augmentations affect the latent space representation. To perform the reduction, we use UMAP [27] to reduce each image from $1024 \times 1024$ pixels to just two components. We flatten each $1024 \times 1024$ image representing a combination of SAR band and augmentation—8736 images in all—into a vector; these vectors are then assembled into an array and passed to the UMAP software. The latent space is generated for all bands and augmented images at once, but we show results by single band before showing the results for all bands together. UMAP takes a "number of neighbors" parameter that determines how the algorithm balances local and global structures in the results [28]; after some experimentation, we set this parameter to 200 as it ensures the preservation of the global structure. We discuss how augmentations affect the resulting latent space representation in Section 3.3.2; first, however, we note how the latent space representation can help illuminate issues with augmentation implementation. The dimensions from the UMAP latent space do not correspond to real-world information. Instead, they represent hidden dimensions UMAP has found that best represent the data in a given lower number of dimensions—analogous to data compression.

### 3.3.1. A Brief Note of Caution

In this section, we note how augmentations can introduce artifacts into the dataset that obscure important features of the original data; in this case, the latent space representation makes this obfuscation clear. Figure 3 presents the two-dimensional (2D) UMAP latent space of the S-band SAR images (see Figure 1 for example images). Each point represents an image; rotation and look angle are identified by marker colors and shapes, respectively. We see that the images tend to cluster in latent space; however, as annotated in the image, the clusters include pairs of rotation angles differing by 90°. This might be surprising based on visual examination; we would not, for example, expect the 90° rotated images (vertical wakes) to cluster with 0° and 180° (horizontal wakes) as they do in Figure 3. Clusters

with rotations differing by 90° (e.g., 30° and 120° or 45° and 135°) have wakes that are orthogonal to each other.

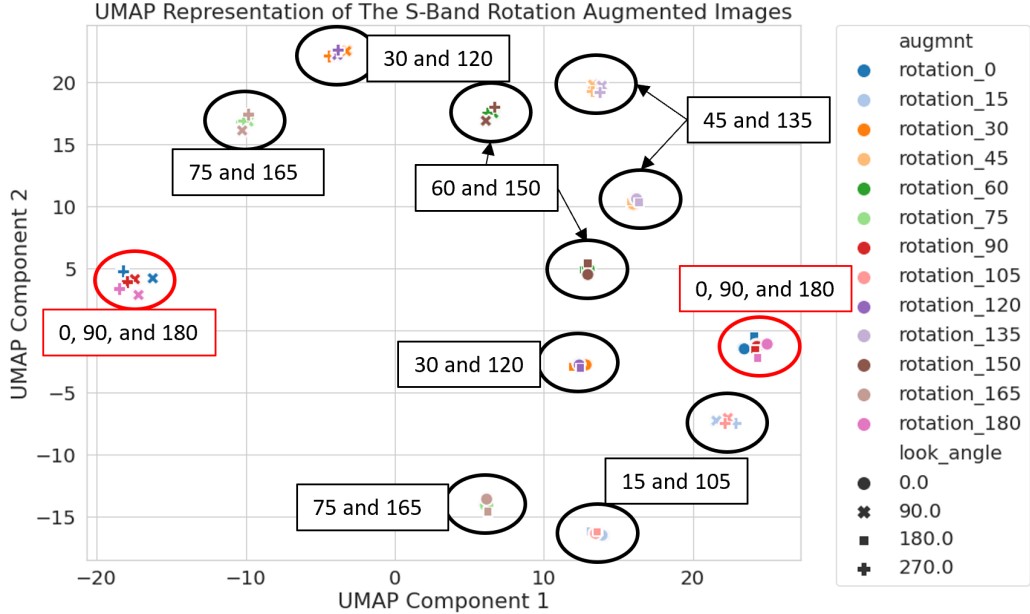

**Figure 3.** S-band latent space with labels for the rotation augmentation angle of each cluster. These labels help to show that clusters are forming based on the difference of the rotation angle from 0° or 90°, revealing inadvertent clustering based on the rotation of a square image.

This behavior, however, is most likely an artifact of the augmentation process. By default, when images are rotated in PyTorch (https://pytorch.org/vision/master/generated/torchvision.transforms.functional.rotate.html#torchvision.transforms.functional.rotate accessed on 4 June 2021), black triangles are formed near the corners of rectangular images; these artifacts are seen in Figure 1. Because the original images are squares, the shapes of these triangles are identical for rotation angles differing by 90°. Rotation angles that are orthogonal will have these triangular edges match (or rather the square outline of the original images will align) despite the wakes not aligning. It appears that UMAP identified these shapes in the latent space representation—rather than learning the effect of augmentations on the wake, adding the augmentation introduced an artificial feature that drowned out the information we wanted UMAP to capture. This just goes to show that understanding the unintentional effects of augmentations on the data is just as important as understanding the desired effects.

PyTorch provides a number of options for "fixing" the corners, such as tiling the image or specifying a fill value. However, since the wakes in our dataset are centered in the images, we simply crop them into circles so the rotation augmentations would not remove any pixel information from the image. With this crop in place, each image then contains the same amount of information, and the rotation only changes the orientation of the wake. We essentially level the playing field by forcing all images to have the same crop regardless of the rotation angle. This way clusters are not formed by the amount of information loss near the corners because all images have the exact same information loss at the corners; see Figure 4 for examples. These circular images are used in all experiments discussed below. For example, in the next section, we demonstrate that the latent space for circular images seems to do a much better job of capturing key features.

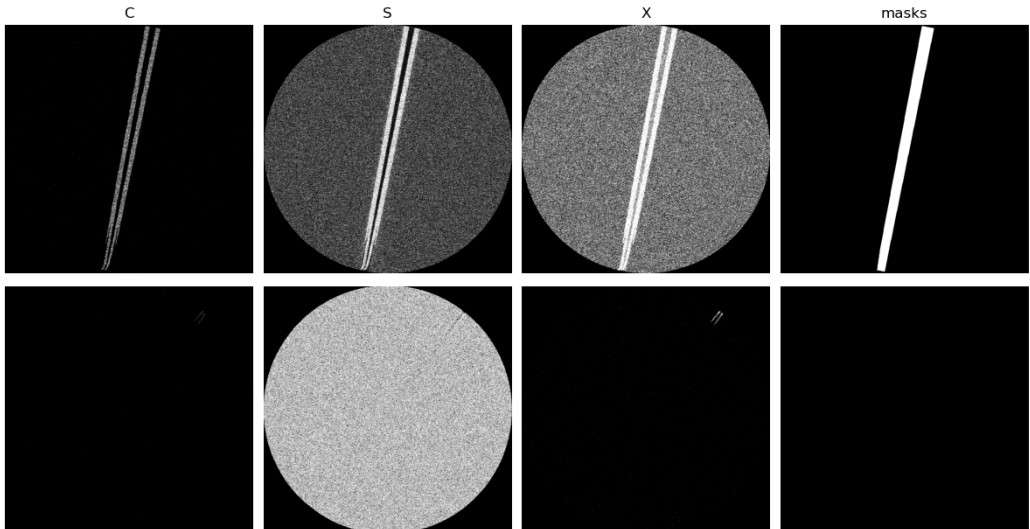

**Figure 4.** Circular cropping is applied to the images so that rotation augmentations do not create triangular clippings that are seen in Figure 1. Any time a rotation is applied to these images, there is no change in the amount or shape of overall information in the image—only the orientation of the wake. Note that there is no persistent wake for the images in the second row, and therefore there is no mask.

### 3.3.2. Effect of Augmentations on Latent Space Representation

In this section, we describe how even a 2D latent space can identify key features in the data that might affect model performance. Figures 5–7 show the latent space representation for the C-, S-, and X-band images, respectively. As in Figure 3, each point represents an image, and rotation and look angles are noted via colors and marker shapes, respectively. Each figure exhibits key features that are worth highlighting.

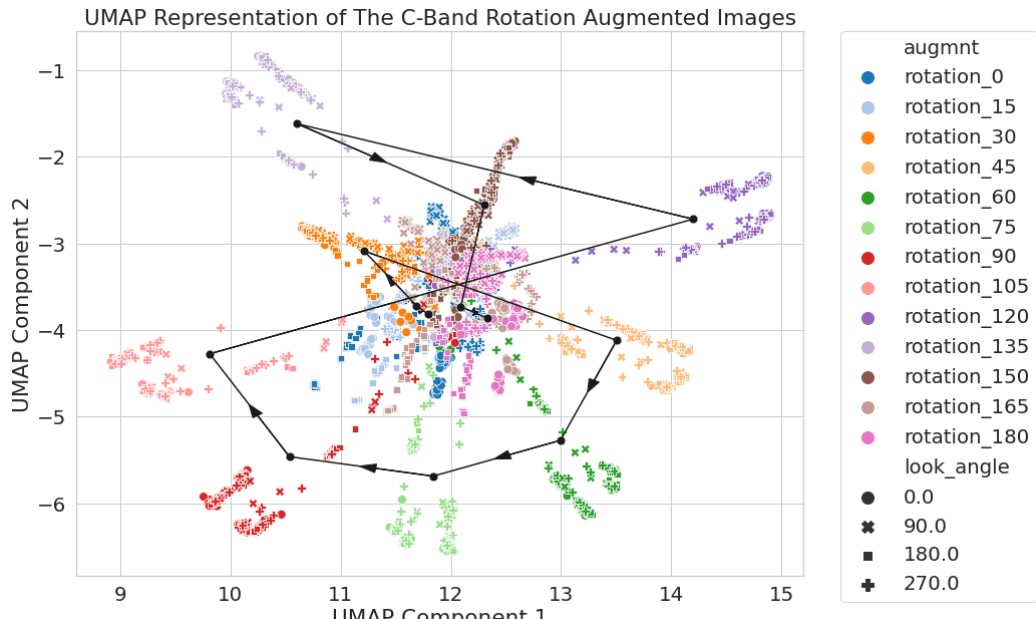

**Figure 5.** C-band only 2D UMAP latent space using circular crop images. Each augmentation set has its mean location pin-pointed with a black dot and connected via a black line with arrows to show the direction of the increasing rotation angle.

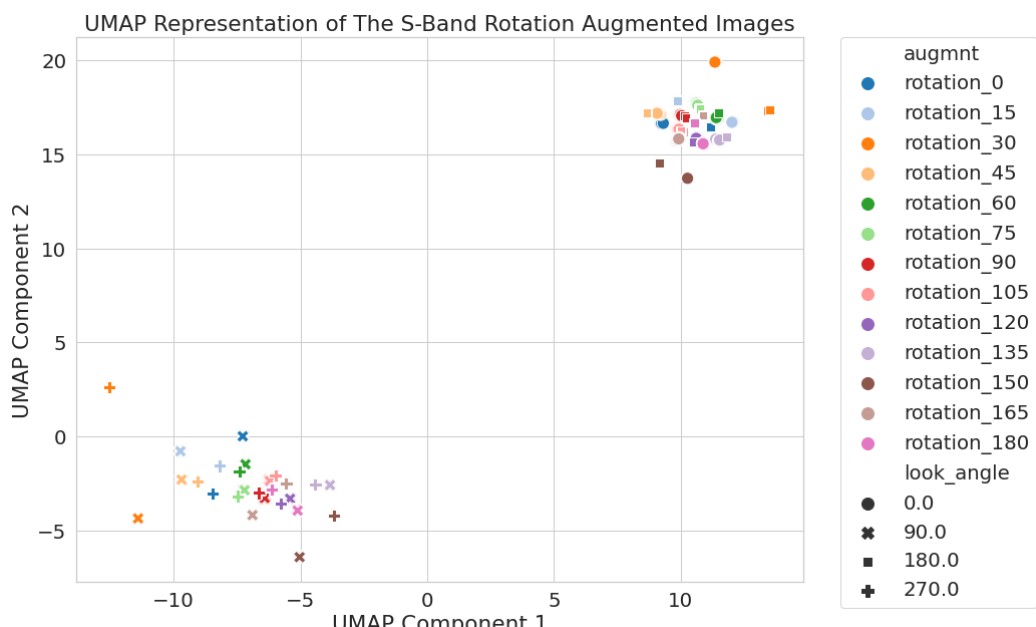

**Figure 6.** S-band only 2D UMAP latent space using circular crop images. All rotation augmentations are used, and the results show two clusters of images based on the look angle, 0° and 180° in the upper right and 90° and 270° are in the lower left.

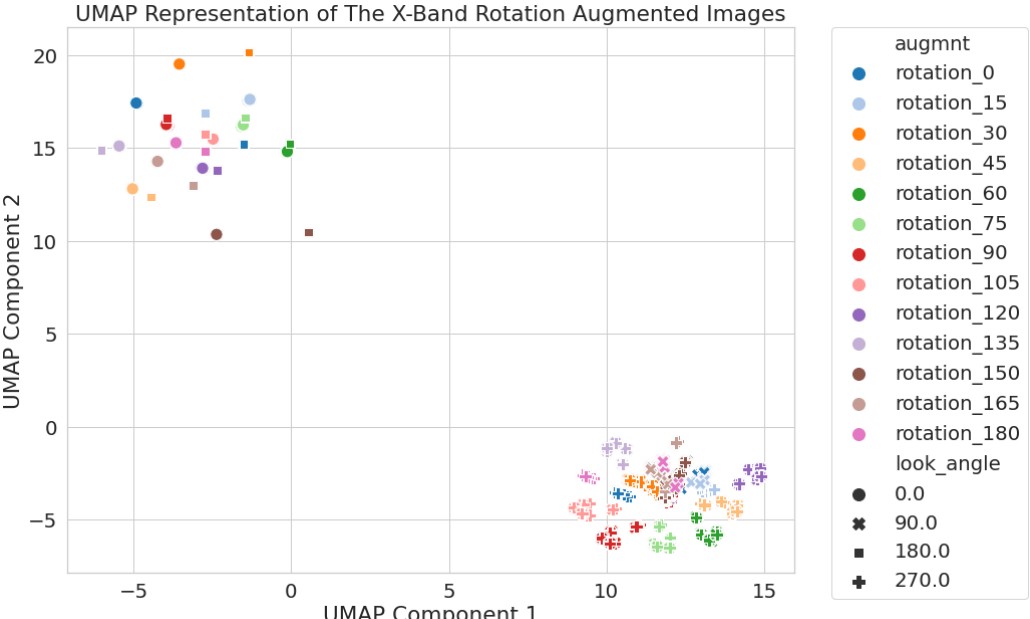

**Figure 7.** X-band only 2D UMAP latent space using circular crop images. All rotation augmentations are used, and the results show two clusters of images based on the look angle, 0° and 180° in the upper left and 90° and 270° are in the lower right. The 90° and 270° look angle images form a pattern similar to C-band (Figure 5), while the 90° and 270° look angle images form a pattern similar to S-band (Figure 6).

For the C-band images (Figure 5), the clusters are complex and do not always have clear and distinct boundaries. The figure shows clear clusters around the outskirts (45° to 135° rotations), with some points falling towards the center, and then a few different augmentation sets that cluster together (0° to 30° and 150° to 180° rotations). To clarify the trend as the rotation angle changes, the mean location of all the points within a set is labeled with a black dot and connected via lines. We can see the general clustering patterns

discussed above: as the sets move from 0°, their location moves outward then zig-zags around before returning back close to 0°. It would also appear that the 90° rotation (the set in the bottom left) is one of the further sets from the center, showing that the latent space representation has the potential to be relevant to the model performance.

The latent space for circular S-band images (Figure 6) shows two clear clusters, contrasting sharply with the 12 clusters for the square images shown in Figure 3, highlighting the effect of removing the clipped corners. These new clusters are determined by the look angle (0° and 180° in the upper right and 90° and 270° in the lower left).

The structure of the latent space for the X-band (Figure 7) is similar, with two large clusters grouped by look angle (0° and 180° in the upper left and 90° and 270° in the lower right). We also see some separation between rotation angles in the lower right cluster, similar to what we saw for the C-band in Figure 5.

The inspection of the images in Figure 8 suggests that the clustering is largely driven by differences in noise level in the background ocean surface between different look angles: look angles of 90° and 270° have significantly less noise in the S- and X- bands than look angles of 0° and 180°. It is also worth noting that the scale of the differences in the latent space is larger for the S- and X-bands than it is for the C-band. The S- and X-band clusters are further from one another than any C-band images are from each other.

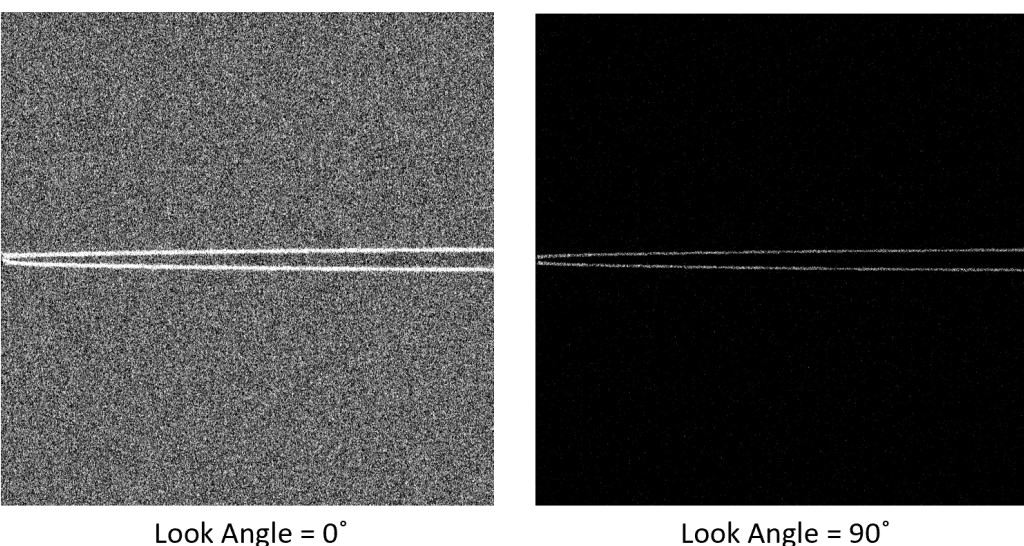

Look Angle = 0°                                             Look Angle = 90°

**Figure 8.** Comparison of the same wake event in the X-Band. On the left is the wake with a look angle of 0°, and on the right is the wake with a look angle of 90°. All other generation parameters are the same for these images. The noise level in the background ocean surface in the left is typical of 0° and 180° look angles, while the one on the right is typical of look angles of 90° and 270°.

Lastly, we present the latent space for all of the images (C-, S-, and X-band) together; the results are shown in Figure 9. We immediately see some interesting features: the 90° and 270° X-band images align right on top of the C-band images, while the S-band and 0° and 180° look angle X-band images form completely separate clusters from the rest of the data.

We expect that a model trained with C-band images would perform well on the 90° and 270° X-band images given that they occupy the same region in the latent space. We also anticipate that a model trained on C-band images would perform poorly for S-band or 0° and 180° X-band images; that is, given the distance between the global clusters, we hypothesize that different models would be required for each, or that careful consideration of the training data would be required to build a model to perform well across all of the clusters. This alignment of images helped to inspire a study to see how well the latent space representation correlates to the performance of the wake classifier models. This study is referred to as the Baseline Latent Space study and is discussed in more detail in Section 4.

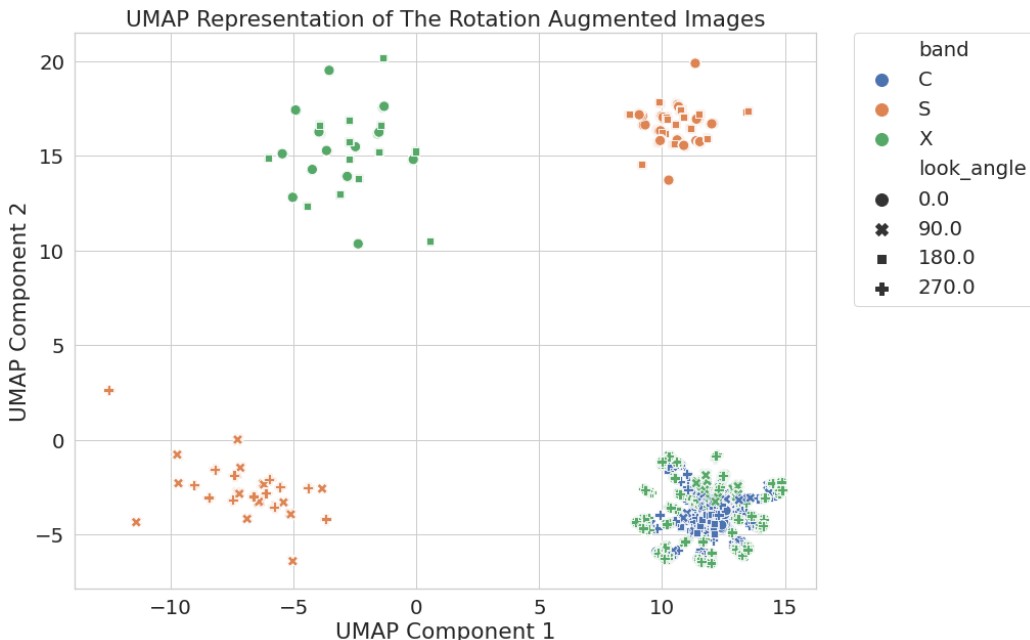

**Figure 9.** Two-dimensional UMAP latent space generated using all rotation-augmented circular crop images. C-band and the 90° and 270° look angle X-band images cluster together in the lower right of the image. S-band and the 0° and 180° X-band images all cluster apart.

## 4. Understanding Model Performance and Latent Space

Now that we can see how augmentations affect the data in the latent space, we test how well the latent space relates to the model performance, if at all. Once we make a connection from augmentations to the data in latent space, then from the data in latent space to model performance, we can assess how best to improve performance via augmentations.

This study focuses on the 2D latent space representation of the data, mostly because it is easiest to visualize and because the entire SAR image dataset forms well-defined clusters in the 2D UMAP space (see Figure 9). One feature of particular interest is how the X-band images with a look angle of 90° and 270° align with the C-band images; this provides the opportunity to train and test models with less concern on manipulating augmentations so we can test how well the latent space relates to model performance. Based on the latent space alignment from Figure 9, we expect those X-band images to test well with a model trained on the C-band images—a proposition that we test in this section.

The models are trained and tested in a different manner here than they were for the sensitivity analysis. We want more testing data to better confirm the alignment in latent space, so we designed four cross-validation folds for testing the data. This split the data (i.e., images) into four folds (i.e., groups) each designed to have the same distribution of `contains_wake`, `look_angle`, and `run_name` as the overall datasets. Different models are trained from scratch using three of the four folds for training and the last fold for testing. The held out fold is rotated so each model is trained and tested four separate times. To ensure each model is trained in the same way, we need to split the data correctly to represent each relevant feature the same way.

Because `contains_wake` is the final label, it is split in a stratified manner to match the overall dataset distribution in each fold. Unfortunately, this results in a fairly imbalanced dataset for each fold. To address this we switch the evaluation metric to the Matthews Correlation Coefficient (MCC) which is better at accounting for imbalanced data than F1 or PR curves [29].

The other features used to split the images are `look_angle` and `run_name`. The look angle, or azimuth angle between the radar direction and the ship direction, proved to be an important feature for the S- and X-band clusters (see Figure 9), so we include it for the folds. We also want to see how the X-band results split by look angle, so we need to make

sure there is an even distribution of look angles for each fold; otherwise, an imbalance could result in poor performance because the model did not see enough examples and not because the data are too difficult to learn. The run name is a combination of the sea state and swell height, which both affect characteristics of the wake formation. Splitting evenly on this feature ensures that the model is trained and tested with all possible wake features in the SAR images.

See Figure 10 for the distribution of the wake generation features for each fold. As we can see `contains_wake`, `look_angle`, and `run_name` features are split in the same way for each fold because these are the most important. The `polarization` feature is binary, so it splits into roughly even ratios for each fold. `inclination_angle` has a great deal more variety, but we do not consider it an important feature for this study, so the distribution there is not of concern.

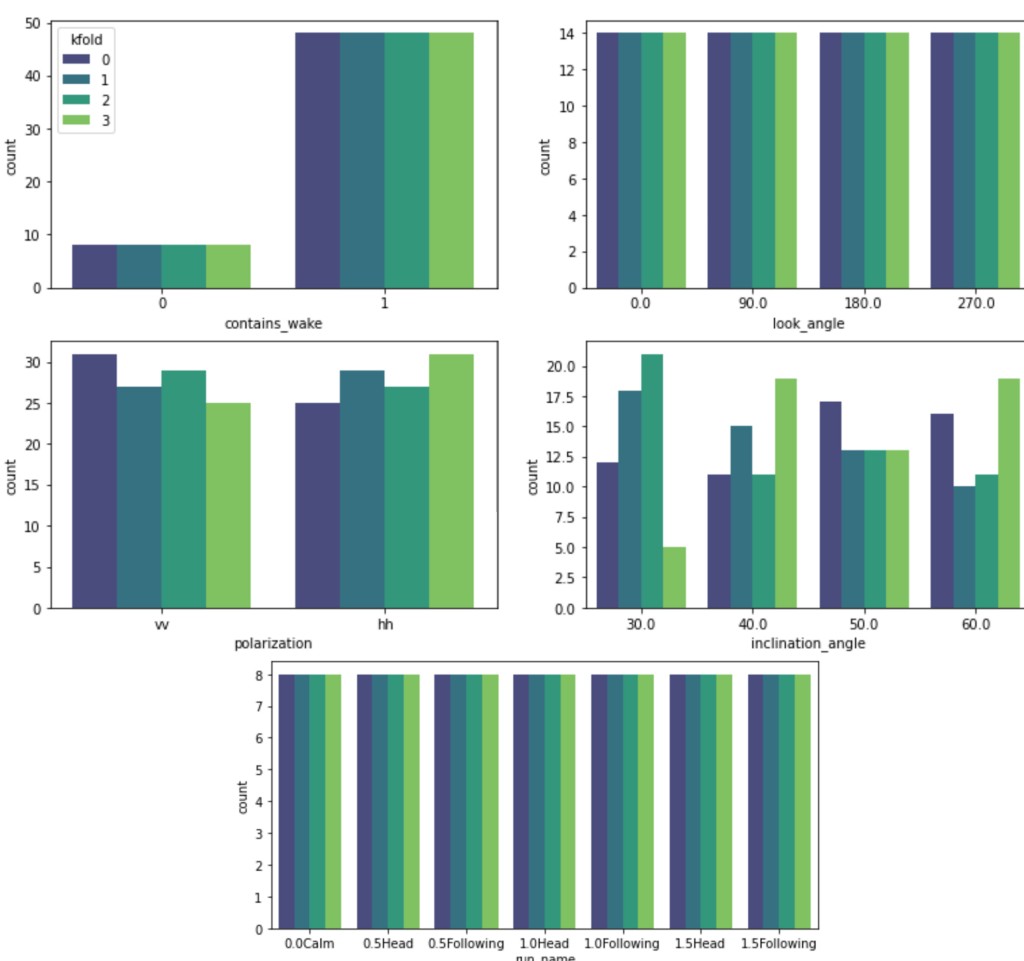

**Figure 10.** Distribution of the metadata features for the four training and testing folds used in the baseline study. The folds are designed to stratify `contains_wake`, `look_angle`, and `run_name` to have matching distributions for each fold.

The rest of the baseline latent space study is straightforward. Models are trained only on the C-band images (the blue dots in Figure 9) and then tested on all bands using the folds discussed. If the latent space representation relates to model performance we expect the models to perform well on the X-band images with look angles of 90° and 270° that align well with the C-band and perform poorly on the rest of the X-band images with look angles of 0° and 180°. The split between X-band look angles creates the control and test groups, as the only difference in those images is the look angle used to generate the SAR imagery. The C- and S-band images act as a sort of control—a control for in domain and a control for out of domain within the latent space. This study is run with four sets of the

data: two sets use no augmentations while the other two use random rotation angles for the images. Within each of those sets the images are either the normal crop images or circular crop images to see how the loss of the information in the corners of the normal crop images might affect the performance of the model.

Results are split by look angle of the SAR image and then aggregated across the test folds. The MCC metric uses a threshold of 0.9 for these results, meaning that a model predicting an output greater than 0.9 is interpreted as a wake in the image, and below that, there is no wake. We chose a high threshold because the models trained with augmented data tend to be overconfident of the presence of a wake, and 0.9 shows good agreement with the C-band results acting as a baseline.

Figure 11 shows the results for the baseline latent space study, where each set of the data returned the same results so only one set is shown. Overall, results match what was hypothesized in the previous section. Each look angle of the C-band performed perfectly (MCC = 1), which is expected because this is a trivial task to train and test on C-band data from the same domain. All the S-band look angles did poorly with MCC = 0; the model in this case predicted wakes present in every image (irrespective of whether there was a wake present or not), which the MCC metric measures as zero (no skill). Finally, for the X-band, we see that it performs perfectly for the 90° and 270° look angles and poorly for the 0° and 180° look angles, just as we expected based on the UMAP latent space image.

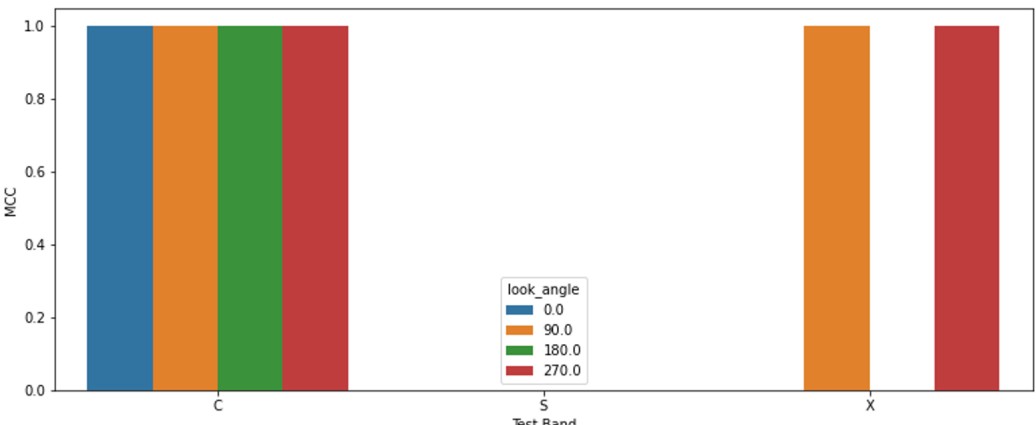

**Figure 11.** Results for the baseline latent space study. This study is run with four sets of the data using a combination of non-rotated or randomly-rotated images and either normal crop image or circular crop images. All sets have matching results, so only one set is presented. The models trained on C-band performed perfectly for C-band and the 90° and 270° look angle X-band images, while it did poorly for S-band and the 0° and 180° look angle X-band images. These results confirm the latent space representation of the data has a relation to the model's performance on that data. The results also convey that latent space is useful regardless of image cropping or applied augmentations. However, cropping and augmentations are consistent between the training and testing images—the only difference is the SAR band.

## 5. Using Latent Space to Improve Model Performance

Now that we see how the latent space representation relates data to the model performance, we test how to improve model performance at the 90° rotation without training the model on that data. Instead, we want to train with as little augmented data as possible and turn to the latent space representation to see which augmentation sets are closest to 90° rotations in the C-, S-, and X-bands. The 90° augmented set serves as a proxy for limited real-world data, and we use the latent space representation of the data to select one augmentation set that boosts model performance at 90°. We focus on the unet model (which combines a U-Net segmentation model and CNN classifier) for this performance study and only show results for that model architecture moving forward.

Before selecting augmentation sets, we run a baseline performance study of naïve models (those trained with no augmentations) tested on 0° and 90° rotated images. Testing on non-rotated images (0°) establishes an expected performance ceiling, while testing on 90° rotated images establishes the expected performance floor. This test is similar to the sensitivity analysis in Section 3.2, but with fewer test sets, and here we run more training and testing iterations with the updated train–test folds discussed in Section 4. Baseline results are shown in the far right column of Figures 12–14.

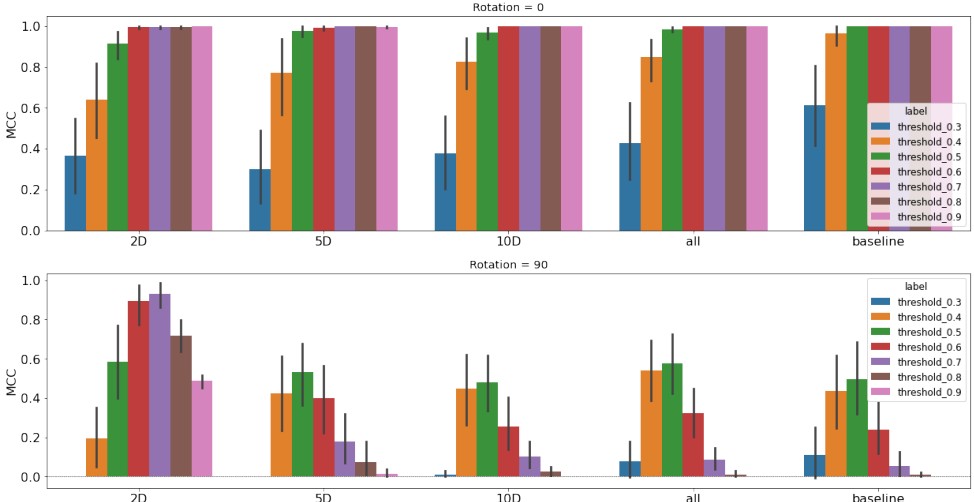

**Figure 12.** Performance study results for C-band using the unet classifier architecture. Each model is trained with an even split of augmented and non-augmented images, then tested on 0° rotated images (**top row**) and 90° rotated images (**bottom row**). We use the MCC metric and a sweep of thresholds to present a profile of performance rather than single metric. Training augmentations are chosen from different latent space representations; moving left to right, they are 2D, 5D, 10D, a combined selection using all three, and then finally the baseline performance is on the far right for comparison.

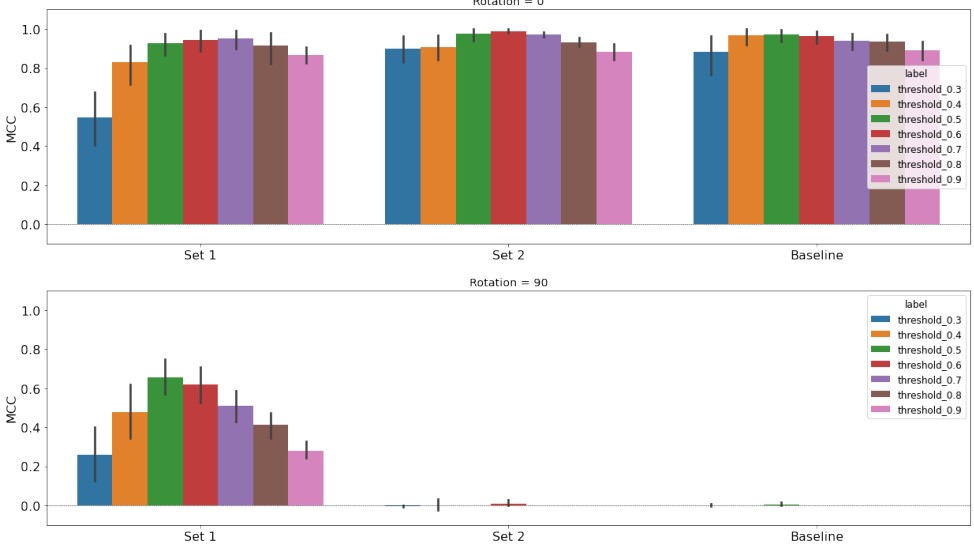

**Figure 13.** Performance study results for S-band using the unet classifier architecture. Each model is trained with an even split of augmented and non-augmented images, then tested on 0° rotated images (**top row**) and 90° rotated images (**bottom row**). We use the MCC metric and a sweep of thresholds to present a profile of performance rather than single metric. Training augmentations are chosen from the 2D latent space representations. Set 1 chooses based on the 90°/270° look angle images, and Set 2 chooses based on the 0°/180° look angle images. The baseline model performance is on the far right for comparison.

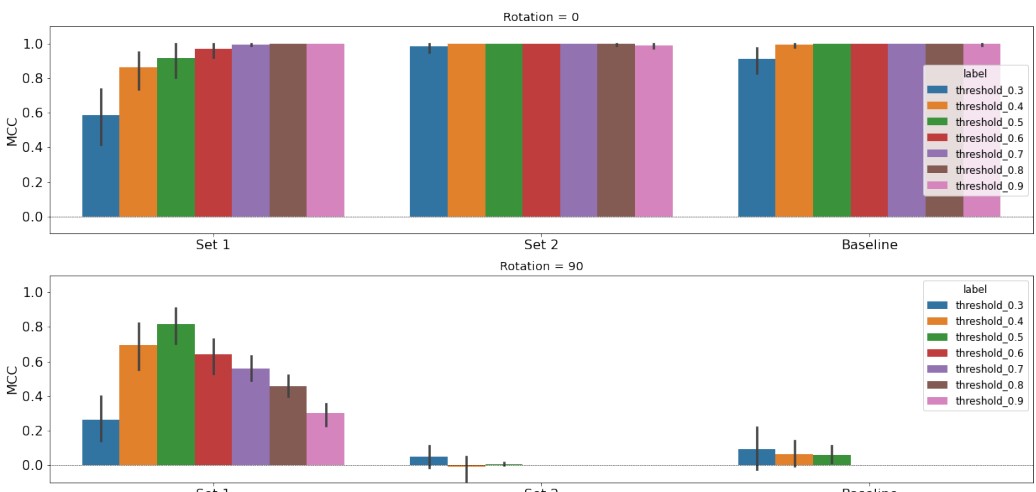

**Figure 14.** Performance study results for X-band using the unet classifier architecture. Each model is trained with an even split of augmented and non-augmented images, then tested on 0° rotated images (**top row**) and 90° rotated images (**bottom row**). We use the MCC metric and a sweep of thresholds to present a profile of performance rather than single metric. Training augmentations are chosen from the 2D latent space representations. Set 1 chooses based on the 90°/270° look angle images, and Set 2 chooses based on the 0°/180° look angle images. The baseline model performance is on the far right for comparison.

### 5.1. Selecting Augmented Sets In Latent Space

To train a model for improved performance in a specific region in the latent space representation, we first need to know which data points are nearest to that region. Figure 5 shows the UMAP components of each augmentation set in a 2D latent space for the C-band images, and Table 2 lists the mean components for each set. We use the UMAP components to calculate the Mahalanobis distance, which helps to characterize the differences between two groups in a multivariate space [30], between 90° and each augmentation set, and present the results in Table 3. The components in Table 2 only contribute to the 2D distances in Table 3. The 5D and 10D distances in Table 3 are calculated using a similar procedure but with the 5D and 10D latent spaces, respectively. The combined dimensions (All) sum distances from the 2D, 5D, and 10D latent spaces. Training sets for each number of UMAP dimensions are listed in Table 4. Because the closest distance to 90° in 10D and All is 0°, we chose the next closest augmentation, which is 15° to be included in the training data. Note that the 5D, 10D, and combination of all dimensions ended up with the same sets for augmentations. Applying some domain knowledge to the problem, we know that 75° is closer to 90° and expect the 2D latent space choice to perform best.

Training datasets are split evenly between non-rotated images (0°) and the chosen augmentation set closest to 90°. This is done based on the cross-validation folds used in the training and testing splits. Each fold is divided evenly between no augmentations and the chosen augmentation in a stratified manner to maintain the same distribution of `contains_wake`, `look_angle`, and `run_name` features as shown in Figure 10. So, each fold has the same representation of augmentations and important features. We run the performance study with five iterations for each of the four folds, just as we did in Section 4.

**Table 2.** Mean 2D UMAP coordinates for the C-band circular crop images from the 2D latent space representation, as shown in Figure 9. For each augmentation set, the rotation angle and coordinates are listed.

| Rotation Angle | Mean UMAP Component 1 | Mean UMAP Component 2 |
|:---:|:---:|:---:|
| 0 | 11.79 | −3.82 |
| 15 | 11.69 | −3.72 |
| 30 | 11.21 | −3.09 |
| 45 | 13.51 | −4.12 |
| 60 | 12.99 | −5.27 |
| 75 | 11.84 | −5.68 |
| 90 | 10.54 | −5.46 |
| 105 | 9.81 | −4.27 |
| 120 | 14.20 | −2.72 |
| 135 | 10.61 | −1.62 |
| 150 | 12.30 | −2.56 |
| 165 | 12.09 | −3.73 |
| 180 | 12.33 | −3.86 |

**Table 3.** Mahalanobis distances between the 90° rotated image set and every other rotated augmentation set in the C-band latent space. Two-dimensional distances are calculated using the coordinates in Table 2, and a similar procedure is used for 5D and 10D representations. The All dimension sums the distances from 2D, 5D, and 10D. The far right column lists the closest rotation angle ($\Phi_{min}$) to 90°.

| Dim. | 0 | 15 | 30 | 45 | 60 | 75 | 90 | 105 | 120 | 135 | 150 | 165 | 180 | $\Phi_{min}$ |
|:---|:---|:---|:---|:---|:---|:---|:---|:---|:---|:---|:---|:---|:---|:---|
| 2D | 1.72 | 1.75 | 2.10 | 2.69 | 2.06 | 1.12 | - | 1.22 | 3.78 | 3.32 | 2.85 | 1.93 | 1.99 | 75 |
| 5D | 2.83 | 2.52 | 4.24 | 3.12 | 3.89 | 4.41 | - | 4.55 | 3.65 | 3.39 | 4.07 | 2.71 | 2.72 | 15 |
| 10D | 3.53 | 4.31 | 4.92 | 4.99 | 4.99 | 4.99 | - | 5.00 | 5.00 | 5.00 | 5.00 | 4.48 | 4.94 | 0 |
| All | 8.09 | 8.58 | 11.27 | 10.80 | 10.94 | 10.51 | - | 10.76 | 12.43 | 11.70 | 11.91 | 9.13 | 9.65 | 0 |

**Table 4.** Training augmentation sets chosen using different latent space representations (UMAP Dimension) for the C-band study. Training augmentations are selected based on the smallest Mahalanobis distance between the 90° rotated images and every other augmentation set. The All UMAP Dimension uses a combination of distances from the 2D, 5D, and 10D latent spaces.

| Band | UMAP Dimensions | Train Augmentations | Test Augmentations |
|:---:|:---:|:---:|:---:|
| C | 2D | 0, 75 | 0, 90 |
| C | 5D | 0, 15 | 0, 90 |
| C | 10D | 0, 15 | 0, 90 |
| C | All | 0, 15 | 0, 90 |

*5.2. C-Band Performance Results*

We chose training sets for C-band based on how close the sets are to 90° in the UMAP latent space for 2, 5, and 10 dimensions and a combination of all dimensions. We present the chosen sets in Table 4 and the results of training models with those sets in Figure 12. The top row of the image shows results when testing on non-rotated images, and the bottom row shows results when testing on the 90° rotated images. The columns for each row are 2D, 5D, 10D, combined dimensions (labelled `all` in the figure), and the baseline.

We calculate results for seven different thresholds (0.3, 0.4, 0.5, 0.6, 0.7, 0.8, 0.9) as a mean value with a 95% Confidence Interval (CI) as a vertical line for each threshold in Figure 12. In order to ensure that the results are not dependent on a single threshold—to see a broader picture of model performance—we present results using several thresholds to calculate the MCC metric.

We tabulate the peak MCC score for each model in Table 5. Performance for the 5D, 10D, and combined all dimensions change little, if at all, from the baseline model

performance. We can clearly see, however, that the 2D latent space option was the only one to have a mean value and 95% CI above the baseline. In fact, the jump is quite significant at 90% improvement from the baseline, showing that the 2D latent space representation is much better at predicting performance based on distance than the others for this wake classification problem. Note that when values are all the same, the 95% CI is N/A because there is no confidence interval (i.e., for a set that contains five values all of which are 1, the mean will be 1 but there is no CI because there is no variance in the data). This occurs for several rows in Table 5 when the results for tests on the 0° rotated images all return 1.0 for their respective threshold and model.

One thing to note about the 5D and all dimension results is that they show a little improvement from the baseline; however, the mean for both fall in the range for the 95% CI of the baseline, and there is a lot of overlap of both their 95% CI with the baseline's. This does not give us much confidence that the results are truly robust. The width of their CI is also concerning because it shows a great deal of variability in the results, and we do not see any improvement of the CI width from the baseline. The 2D results, however, are well outside the 95% CI of the baseline, have no CI overlap with the baseline, and show a narrowing of the CI, meaning those result are more consistent.

These results also show that improvements are possible for a moderately performing model when selecting augmentations using information from the latent space. Using the correct latent space representation, however, does matter, and when working on new datasets, it might take some iterations or domain knowledge to determine the best options for generating that representation.

**Table 5.** C-band results tabulating unet model training data, testing data, the peak MCC performance with 95% confidence interval (CI), and the corresponding threshold.

| Dimensions | Training Rotation | Testing Rotation | MCC | 95% CI | Threshold |
|---|---|---|---|---|---|
| 2 | 0, 75 | 0 | 1.0 | N/A | 0.9 |
| 5 | 0, 15 | 0 | 1.0 | N/A | 0.7 |
| 10 | 0, 15 | 0 | 1.0 | N/A | 0.6 |
| All | 0, 15 | 0 | 1.0 | N/A | 0.6 |
| Baseline | 0 | 0 | 1.0 | N/A | 0.5 |
| 2 | 0, 75 | 90 | 0.93 | 0.069 | 0.7 |
| 5 | 0, 15 | 90 | 0.53 | 0.17 | 0.5 |
| 10 | 0, 15 | 90 | 0.48 | 0.16 | 0.5 |
| All | 0, 15 | 90 | 0.58 | 0.17 | 0.5 |
| Baseline | 0 | 90 | 0.49 | 0.19 | 0.5 |

*5.3. S- and X-Band Results*

The S- and X-band studies were conducted after the C-band study in the previous section. After finding that the 2D representation was the best performing latent space to use for this process as described in Section 5.2, we use only the 2D latent space for the S- and X-band studies. These bands are also dependent on the look angle of the SAR image, forming clusters based on look angles of 90° and 270° or 0° and 180° (see Figures 6 and 7). Therefore, we broke the selection down by the look angle cluster and then picked training sets based their closeness to the 90° rotated images within each cluster. This results in two options for each band in the 2D UMAP latent space, which are shown in Table 6.

Unlike the C-band, the S- and X-band baselines have no skill with the 90° rotated images, so we can surmise that these bands are more difficult for the model to use. We present the results for the S-band models in Figure 13 with tabulated results in Table 7 and the X-band models in Figure 14 with tabulated results in Table 8. The models were trained with images rotated at 0° and 75° (Set 1), 0° and 180° (Set 2), or only 0° (baseline).

**Table 6.** Training augmentation sets chosen using different look angles (0° and 75° for Set 1, and 0° and 180° for Set 2) for the S- and X-band studies. Training augmentations are selected based on the smallest Mahalanobis distance between the 90° rotated images and every other augmentation set.

| Band | Set | UMAP Dimensions | Train Augmentations | Test Augmentations | Look Angle |
|------|-----|-----------------|---------------------|--------------------|------------|
| S | 1 | 2D | 0, 75 | 0, 90 | 90/270 |
| S | 2 | 2D | 0, 180 | 0, 90 | 0/180 |
| X | 1 | 2D | 0, 75 | 0, 90 | 90/270 |
| X | 2 | 2D | 0, 180 | 0, 90 | 0/180 |

**Table 7.** S-band results tabulating unet model training data, testing data, the peak MCC performance with 95% confidence interval (CI), and the corresponding threshold.

| Set | Training Rotation | Testing Rotation | MCC | 95% CI | Threshold |
|-----|-------------------|------------------|-----|--------|-----------|
| 1 | 0, 75 | 0 | 0.95 | 0.054 | 0.7 |
| 2 | 0, 180 | 0 | 0.99 | 0.011 | 0.6 |
| Baseline | 0 | 0 | 0.97 | 0.056 | 0.4 |
| 1 | 0, 75 | 90 | 0.66 | 0.092 | 0.5 |
| 2 | 0, 180 | 90 | 0.010 | 0.021 | 0.6 |
| Baseline | 0 | 90 | 0.0057 | 0.012 | 0.5 |

**Table 8.** X-band results tabulating unet model training data, testing data, the peak MCC performance with 95% confidence interval (CI), and the corresponding threshold.

| Set | Training Rotation | Testing Rotation | MCC | 95% CI | Threshold |
|-----|-------------------|------------------|-----|--------|-----------|
| 1 | 0, 75 | 0 | 1.0 | N/A | 0.8 |
| 2 | 0, 180 | 0 | 1.0 | N/A | 0.4 |
| Baseline | 0 | 0 | 1.0 | N/A | 0.5 |
| 1 | 0, 75 | 90 | 0.82 | 0.11 | 0.5 |
| 2 | 0, 180 | 90 | 0.048 | 0.071 | 0.3 |
| Baseline | 0 | 90 | 0.093 | 0.13 | 0.3 |

For both bands, we see after testing on the 90° images (bottom row) that Set 1 had a significant improvement and Set 2 had none. The X-band shows slightly more improvement than the S-band model, resulting in a peak MCC of 0.82, whereas the S-band has a peak of 0.66. What is also important to note is the slight degradation of Set 1's results on the non-rotated test images (top row). We have made vast improvements in the desired augmentation (90°) while sacrificing some performance on the original dataset. The C-band results had a similar, albeit much more subtle, effect for the 2D results (see Figure 12 and Table 5).

Both X- and S-band models show improvement with Set 1; however, there is also a disparity between the threshold for optimal performance when testing on the 0° rotated images (threshold = 0.7 and 0.8) and 90° rotated images (threshold = 0.5) (see the peak thresholds of Set 1 for Figures 13 and 14). Not only do we sacrifice some performance on the original dataset, but this forces us to choose a threshold to balance performance in these two domains if we were to implement the models in a real-world scenario.

The lack of any performance increase for Set 2 is striking in that the same approach as Set 1 yields the differing performance that we see in Figures 13 and 14. The deep learning models and latent space representations use all the same data, so we conclude that the latent space is affected by features that the deep learning models appear to be robust against. While the latent space representation can tell us a great deal about our data, using that information is no guarantee of model improvement.

The S- and X-band results show us how well the latent space can help improve a model when there is a lack of performance in that model on data that are completely outside its original operating envelope. However, this requires some careful considerations and knowledge of the dataset. The latent space can pick up important data features that affect model performance, but it can also pick up other features that do not affect the model.

## 6. Discussion

Data augmentations not only alleviate the problem of limited dataset sizes for training deep learning models but also help to increase model performance by altering the data in meaningful ways. We used rotation augmentations to alter ship wake SAR images in a single direction to any orientation in a matter of seconds instead of hours re-running simulations. This is crucial in situations where data are limited and gathering more is not feasible, but we can augment to expand the data size and domain in ways to boost model performance. There are limitations, however, to what augmentations can do. For our wake images, we have to make sure the labels and underlying physics are not altered, but this is not a large concern with the rotation augmentations. Augmentations are not a panacea but a tool to help make available data more useful and need to be used with domain knowledge.

The effects of augmentations on the naive models are striking but not surprising. The sensitivity analysis (Section 3.2) revealed the 90° rotation augmentations are the worst performing region of the data domain across all models and bands. The structure of the augmented data was revealed through a latent space generated by the UMAP technique (Section 3.3). Looking at the C-band latent space in Figure 5, 0° and 180° rotation angles group at the center and had the highest scores with the naive models. As the rotation angle moves from 0° to 180°, the data depart from the center, and performance drops. The proximity in latent space to the training data correlated with the performance of the C-band model.

When looking at the S- and X-band data (Figures 6 and 7), the latent space also reveals a structure in the generative parameters of the data—specifically the look angle used. There is only a vague pattern in the S- and X-band data compared to C-, but the S-band latent space exposed an issue with the way augmentations altered the images by clipping corners. It showed the sensitivity of the latent space to unintended changes and how latent space can uncover patterns we did not know to look for.

The global pattern of all bands in the latent space revealed a significant section of the X-band exists in the same space as the C-band, and we showed that a C-band model performs well on that X-band data in Section 4. As expected, all other images outside that space performed poorly. Ideally, we can use the latent space to discern an operating envelope of any model to predict model performance on certain data, or more proactively, we could design a training set to encompass as much of the data domain as possible, maintaining performance, while minimizing its size.

The latent space representation of the wake images, including augmented images, is meaningful for model performance. The results in Section 4 show that the relationship between the X-band and C-band data is an indicator of performance for the X-band images with the C-band trained model. While the study was fairly simplistic, it gives us confidence that the latent space can be used to determine how we can best train a model for the desired performance. This does require a deeper understanding of the data domain—we know we have to split X-band based on look angle, but we can reduce the training dataset and combine a model for C-band and the 90° and 270° look angle X-band images.

As disheartening as it is to see that most of the options for the C-band in the performance study (Section 5.2) did not work out, it is exciting to see how well the 2D latent space improved the baseline model's performance. This is promising in that the latent space is useful for improving model performance with limited augmented data. In addition, we see some similar results for S- and X-band models using Set 1 (Section 5.3). The 2D latent space is likely the best option here because (1) it encodes the most amount of information into

as few components as possible and (2) adding more dimensions increases the number of components (i.e., noise) affecting the distance calculations. It is possible that the distance calculation is losing the signal of important components as more latent space dimensions are added.

The S- and X-band results show that important features in the latent space, such as the look angle, can have a significant effect on model performance. Depending on the look angle cluster used in Section 5.3, there is either significant improvement (for 90° and 270° look angles in Set 1) or none at all (for 0° and 180° look angles in Set 2). We see that latent space can be important for understanding model performance, but at the same time, it is subject to other effects that the models are not. This emphasizes the importance of understanding your data, models, and latent space in order to design a training set that improves performance for your desired goal. If we apply our domain knowledge of angles, the results are not too surprising. We know that 75° is closer to 90° than 15° or 180°. Combining domain knowledge and latent space is the best way to move forward. Some experimentation, however, is required to see how well a latent space maps to model performance. There is not a direct one-to-one correlation, and any relationship is likely to vary from dataset to dataset, model to model, and latent space to latent space.

## 7. Conclusions

In this paper, we demonstrated that meaningful improvements can be made to a deep learning model's performance on a limited size dataset by understanding the latent space and effect of augmentations to the data. We conducted a novel experiment to connect the unet classifier's performance to the data in latent space and to augment data with meaningful alterations for improving model performance. This improvement is critical in situations where data are limited, as augmentations not only expand size but also the data domain. There are limitations, however, concerning how much alteration is meaningful in a physical sense. Augmentations are a tool to help make available data more useful and need careful application.

Latent space is meaningful to performance in certain circumstances, and we demonstrated this with the baseline study showing latent space predicted performance (Section 4) and with the C-band performance study using the 2D UMAP latent space (Section 5.2). However, higher dimensional latent spaces proved ineffective for the C-band, resulting in models that were no better than the baseline.

The S- and X-band data required training sets to be selected that compounded augmentation and look angle. Results were mixed for these bands: some models exhibited significant improvement from the baseline, and others showed none (Section 5.3). These results show the importance of understanding the latent space, having domain knowledge, and experimenting to make it all work for the data selection process.

Latent space representation can lead to improved model performance, but with the trade-off of requiring deeper model and data understanding. We believe this level of understanding should be a prerequisite for training deep learning models regardless of your task. The procedures laid out in this paper are not the end-all-be-all of training with limited data, but they can help to orient smaller sets of experiments to help improve models in areas with little data.

**Author Contributions:** D.S.: conceptualization, methodology, software, validation, analysis, investigation, visualization, writing—original draft preparation. E.H.: software, data curation, writing—original draft preparation. J.K. (Justin Krometis): writing—review and editing. J.K. (Justin Kauffman): writing—review and editing. L.F.: supervision, funding acquisition, writing—review and editing. All authors have read and agreed to the published version of the manuscript.

**Funding:** This research was funded by Alion Science and Technology in support of the Scientific Test and Analysis Techniques Center of Excellence (STAT COE) and the Homeland Security Community of Best Practices (HS CoBP), Contract: FA8075-14-D-0019; DSC3133-03 Virginia Tech. Approved for Public Release, Distribution Unlimited.

**Institutional Review Board Statement:** Not applicable.

**Informed Consent Statement:** Not applicable.

**Data Availability Statement:** The data presented in this study are openly available on GitHub at https://github.com/dssobien/wake_data_augmentation (accessed on 12 May 2022).

**Acknowledgments:** This work was supported in part by high-performance computer time and resources from the DoD High Performance Computing Modernization Program.

**Conflicts of Interest:** The authors declare no conflict of interest.

## Abbreviations

The following abbreviations are used in this manuscript:

| | |
|---|---|
| ASC | Attributed Scattering Centers |
| ATR | Automatic Target Recognition |
| CAD | Computer Aided Design |
| CFD | Computational Fluid Dynamics |
| CNN | Convolutional Neural Network |
| CST | Computer Simulation Technology |
| csv | Comma-Separated Values |
| EO | Electro-Optical |
| ERIM | Environmental Research Institute of Michigan |
| GAN | Generative Adversarial Network |
| MCC | Matthews Correlation Coefficient |
| MDPI | Multidisciplinary Digital Publishing Institute |
| MRI | Magnetic Resonance Imaging |
| PCA | Principal Component Analysis |
| PR | Precision-Recall (Curve) |
| PR AUC | Area Under the Precision-Recall Curve |
| ROC | Receiver Operating Characteristic |
| SAR | Synthetic Aperture Radar |
| t-SNE | T-distributed Stochastic Neighbor Embedding |
| UMAP | Uniform Manifold Approximation and Projection |
| UUID | Universally Unique Identifier |

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
