# Peer review of "Improving Deep Learning for Maritime Remote Sensing through Data Augmentation and Latent Space"

_make, doi:10.3390/make4030031_

Round 1

Reviewer 1 Report

The authors have addressed using the data augmentation (DA) technique on the simulated SAR images of ship wakes. Apart from the conventional expectation of DA to alleviate the problem of limited dataset sizes for training deep learning models, the aspect of increasing model performance by altering the data in meaningful ways is very interesting. The paper seems to be sound from the technical point of view, however, I have the following points to improve the quality of the paper:

Regarding DA (specifically image rotation) it is recommended that the authors highlight the differences in SAR and optical images in the last paragraph of 2.1., similar to the following paragraph from A Survey on the Applications of Convolutional Neural Networks for Synthetic Aperture Radar: Recent Advances:

In EO domain, data augmentation technique typically consists of scaling, cropping, padding, rotation, flipping, translation and so on. However, SAR data augmentation must be tied to the sensor itself meaning that rotation and flipping will not work as it does in EO domain because of the shadowing issues.

Moreover, although Section 2 (background) is well structured (by having 4 subsections), the literature on DA and the use of simulated SAR images is not comprehensive enough, and more studies are needed to cite. As a few examples extracted from the above-mentioned survey: 

-          Ding et al. [0] proposed a CNN for the SAR-ATR task with the emphasis on data augmentation techniques such as adding speckle noise, translation and pose synthesis.

-          Kwak et al. [1] have used speckle noise addition to make their CNN robust for SAR-ATR tasks.

-          Du et al. [2] proposed a displacement- and rotation-insensitive CNN for data augmentation in MSTAR dataset. 

-          Lv and Liu [3] used the concept of attributed scattering centers (ASC) for data augmentation in SAR images.

Instead, some studies followed different approaches such as transferring knowledge from simulated SAR data, unlabeled SAR images, and so on:

-          Hansen et al. [4] showed that a CNN, pretrained on simulated data outperforms the one that is trained only on real data, especially when the labeled real data is not sufficient. They transferred knowledge from a simulated SAR dataset and fine-tuned it by using MSTAR dataset.

-          Similarly, Wang et al. [5] utilized transfer learning between simulated SAR data and real SAR data to solve the problem of insufficient training samples. They used adversarial domain adaptation to handle the problem of domains shift between the source and the target datasets.

[0] Ding, B. Chen, H. Liu, M. Huang, "Convolutional neural network with data augmentation for SAR target recognition", IEEE Geosci. Remote Sens. Lett., vol. 13, no. 3, pp. 364-368, Mar. 2016.

[1] Y. Kwak, W.-J. Song, and S.-E. Kim, “Speckle-noise-invariant convolutional neural network for SAR target recognition,” IEEE Geosci. Remote Sens. Lett., vol. 16, no. 4, pp. 549–553, Apr. 2019.

[2] K. Du, Y. Deng, R. Wang, T. Zhao, N. Li, “SAR ATR based on  displacement- and rotation-insensitive CNN”, Remote Sens. Lett., vol. 7, no. 9, pp. 895-904, 2016.

[3] J. Lv and Y. Liu, “Data augmentation based on attributed scattering  centers to train robust CNN for SAR ATR,” IEEE Access, vol. 7, pp.  25459–25473, 2019

[4]D. Malmgren-Hansen, A. Kusk, J. Dall, A. A. Nielsen, R. Engholm and  H. Skriver, “Improving SAR automatic target recognition models with  transfer learning from simulated data,” IEEE Geosci. Remote Sens. Lett., vol. 14, no. 9, pp. 1484-1488, Sep. 2017

[5] K. Wang, G. Zhang, H. Leung, “SAR Target Recognition Based on Cross-Domain and Cross-Task Transfer Learning,” IEEE Access, vol. 7, pp. 153391-153399, 2019.

Reviewer 2 Report

Sobien and co-authors investigated the topic of the improvement of the training process in the context of wake retrieval in SAR data using the Uniform Manifold Approximation and Projection (UMAP). UMAP is a dimensionality reduction technique used to analyze the effects of augmentations on the deep learning process.

The article is well written, treats an actual issue and analyzes in detail the use of the UMAP technique applied to the analysis of data augmented simulated datasets, its limits and its potential.

I really appreciated the authors' efforts to find in the UMAP analysis an inference of the performance of the network on real data.

Although the authors stated that latent space representation can tell us a lot about our data, but using that information is no guarantee of model improvement, I really enjoyed reading this work. 

Despite the fact that UMAP analysis is not the panacea for all our problems, it is definitely a step forward to try to understand in a more controlled way the complex mechanism of deep learning.

There are a few minor aspects that might be improved which are listed below.

Fig. 1: the mask in the second  row is not visible

3.3.1 When the authors wrote about artifacts introduced by UMAP they discussed triangles at the edges of the image generated by the rotation of the original images. If I understand correctly, they argue that these triangles at the edges are misinterpreted by the UMAP representation as fake-wakes shapes (or at least as part of fake-wakes shapes) in the data, since the geometric structure of a wake is made up of two lateral bright triangles and an inner dark-gray triangle.  

It is not clear to me how the authors are sure they can identify in these extra features in the image the clusterization issue of the images rotated of 0°, 180° and 90° shown in Fig. 3.

A further observation. Why not put NAN values in the triangles generated by the rotation?

3.3.2 

Fig. 4: the mask in the second  row is not visible

The choice of colours in the UMAP representations is very hard to interpret. In particular, I find Fig. 5 critical, being many classes to identify.

What kind of “inspection of the images”  the authors are considering when they conclude that “clustering is largely driven by differences in noise level between different look angles” and how do they read this in Fig. 6 and 7?

4 Please explain better the design of the training and validation folder (p. 10  line 0-10)

5.2 The authors wrote, “We calculate results for several thresholds as a mean value with a 95% Confidence Interval (CI) as a vertical line for each threshold”.

Can you give a quantitative assessment? How many thresholds?

Why in Table 5 in the column 95% CI are all N/A for testing rotation 00?

Reviewer 3 Report

1.      Basically, the SAR systems always work at side-look mode, that is to say, the Look angle 0 and 180 degree defined in Table 1 are not reasonable. 90 and 270 may denote right-look and left-look separately, which are more feasible to some extent. 

2.      Please add the math definition of PR AUC metric. 

3. The motivation of creating the latent space is not described explicitly. Besides, the main method adopted in the manuscript is UMAP software, please make the contribution of the manuscript more clearly. 

4. The captions in Figure 3/5/6 are blurred, please improve it.

Reviewer 4 Report

This manuscript is focused on explaining the influence of rotation data augmentation on model performance through UMAP and how best to employ data augmentations in a more effective manner.

There are some flaws to be improved

1. In general, the mentioned “rotation” refers to geometrical rotation of images in deep learning rather than the different look angels in SAR imaging. But there is obvious difference for different look angels SAR image, such as layover, shade and so on. Do you consider the image grey variation when rotating the SAR image?

2. If the datasets are trained with one wake direction, such as horizontal or vertical, the direction will be considered as an important feature for the model. Therefore, different rotation angles show different performance. Do you adopt different wake angels in the figure2? You know, there are different real wake SAR images at different angels.

3. In section 2.3, the simulated parameters are not comprehensive. You should add wind, hull shape and size, speed and heading which affect the visibility of ship wakes in SAR images.

4. Why is there no sea clutter on some images in the figure1, such as the first column on the left?

5. Please add the explanations of x and y coordinate axis of 2D UMAP latent space.

Round 2

Reviewer 3 Report

Please have a through review about the "Look angle" of a SAR system, then the reasonable values could be clarified. 

Reviewer 4 Report

No more coments

Author Response

We thank the reviewer for their previous comments and acceptance of our corrections.